# Lower Plasma Lactate Concentrations After Training Support the Hypothesis of Improved Metabolic Flexibility in Male Long-Term Selected Marathon Mice Compared to Unselected Controls

**DOI:** 10.3390/cells13242123

**Published:** 2024-12-21

**Authors:** Julia Brenmoehl, Zianka Meyer, Christina Walz, Daniela Ohde, Andreas Hoeflich

**Affiliations:** Workgroup Endocrinology of Farm Animals, Research Institute for Farm Animal Biology (FBN), 18196 Dummerstorf, Germanyhoeflich@fbn-dummerstorf.de (A.H.)

**Keywords:** marathon mice DUhTP, metabolic cages, respiratory quotient, lactate, fat oxidation rate, treadmill training, metabolic flexibility

## Abstract

Metabolic flexibility describes the capability to switch between oxidative fuels depending on their availability during diet or exercise. In a previous study, we demonstrated that in response to training, marathon (DUhTP) mice, paternally selected for high treadmill performance, are metabolically more flexible than unselected control (DUC) mice. Since exercise-associated metabolic flexibility can be assessed by indirect calorimetry or partially by circulating lactate concentrations, we investigated these parameters in DUhTP and DUC mice. Therefore, males of both lines completed a three-week high-speed treadmill training or were physically inactive (sedentary) before being placed in a metabolic cage for three days (one day of acclimatization, two days with monitoring), measuring CO_2_ and O_2_ to calculate respiratory quotient (RQ) and fatty acid oxidation (FATox). Circulating blood lactate concentrations were determined. Training resulted in a lower RQ in DUhTP and an increased RQ in DUC mice compared to their sedentary counterparts. Increased FATox rates and lower lactate concentrations were observed in exercised DUhTP but not in DUC mice, indicating a shift to oxidative metabolism in DUhTP and a glycolytic one in DUC mice. Therefore, improved metabolic flexibility in DUhTP mice is verifiable up to three days after training.

## 1. Introduction

Metabolic flexibility generally refers to the capacity to switch between oxidative fuels depending on fuel availability (lipids or carbohydrates) [1], e.g., during the transition from fasting to a state of satiety [2]. Postprandial, the mitochondria oxidize a mixture of fat and carbohydrates and, following fasting (postabsorptive), mainly fats [3], with fatty acid flux in the postprandial circulation being regulated by adipose tissue and glucose flux by the liver and, to a lesser extent, skeletal muscle [4]. In healthy, lean individuals, the intake of a high-carbohydrate diet after fasting leads to a slight increase in blood insulin levels and a strong shift from fatty acid to glucose oxidation while blood glucose levels remain constant [3]. In contrast, a high-fat diet increases fatty acid oxidation at the expense of glucose [5].

Metabolically inflexible subjects exhibit impaired fuel switching, which causes persistent oxidation of all three major fuels (fats, carbohydrates, and amino acids) and associated mitochondrial overload, which has been linked to metabolic disorders such as obesity, insulin resistance, and type 2 diabetes [3,6]. Diminished metabolic flexibility is manifested by reduced fat oxidation during nocturnal fasting and decreased insulin-stimulated carbohydrate oxidation, and this inability to regulate lipolytic and glycolytic processes is associated with obesity [7,8]. After high glucose availability, i.e., following an insulin–glucose infusion after fasting [7], metabolically inflexible individuals show impaired suppression of lipolysis, which is associated with increased concentrations of free fatty acids in the bloodstream [9]. These free fatty acids, in turn, reduce the rate of glucose utilization by inhibiting glycogen synthesis and glucose oxidation [9] despite high glucose availability. The administration of a high-fat diet in subjects with impaired metabolic flexibility leads to hardly any change in the expression of genes involved in fatty acid oxidation in the skeletal muscles, whereas metabolically flexible individuals display increased transcription [10]. Excessed fatty acids accumulate in fat depots and as ectopic fat, such as in skeletal muscle tissue. Intramyocellular-accumulated lipid concentrations correlate with insulin resistance and alter insulin signaling [11].

However, metabolic flexibility can be improved by calorie restriction, substantial weight loss, or exercise. A weight loss of approximately 15 kg in overweight subjects through calorie restriction enhances the ability to utilize glucose in response to insulin-stimulated conditions [12]. Ten days of aerobic endurance training can increase fat oxidation in the skeletal muscle of overweight probants in response to short-term excursions with dietary fat [13].

A certain metabolic flexibility is required during physical exercise to efficiently coordinate the enormous increase in energy demand with fuel availability. Exercise training alters, in turn, fuel storage and availability in skeletal muscle epigenetically [14,15], transcriptionally [16,17,18], and translationally [19,20] to “flexibly” meet the altered energy demands of each workout [2]. Recreational athletes have lower metabolic flexibility and inefficient anaerobic contributions (glycolytic and phosphagen systems) and, thus, lower energetic performance than advanced or professional athletes due to their mitochondrial function and cardiovascular fitness [21]. A high degree of metabolic flexibility is reflected in the physiological adaptability that enables an appropriate response to the increasing energy demand and availability of fuels during exercise in terms of substrate recognition, transport, utilization, and storage [2,22]. In connection with the increasing energy demand during regular exercise, the oxidation of metabolically flexible individuals and the transcription of genes that regulate carbohydrate (CHOox) or fat oxidation (FATox) in skeletal muscle increases depending on the available fuels. Transcription factors such as peroxisome proliferator-activated receptor gamma coactivator 1-alpha (PGC1α) [8] or transcription factor EB (TFEB) [23] play an important role in this process. They directly or indirectly regulate nuclear and mitochondrial-encoded genes that are required for contractile and metabolic adaptations in skeletal muscle [22,24], oxidative phosphorylation, lipid transport, and oxidation [25], glucose uptake and metabolism, glycogen production and accumulation, and promote lipid metabolism and mitochondrial biogenesis in muscle to ensure energy production during exercise [23].

Recently, we raised the hypothesis that the long-term-selected mouse line DUhTP is characterized by improved metabolic flexibility [17]. Although this mouse model was paternally long-term selected for high treadmill performance, these mice were distinguished by above-average running endurance without any exercise intervention [26]. After a three-week training trial, we found higher Tfeb and Ppargc1a (PGC1α) expression than in unselected control mice (DUC) originating from the same genetic background [17]. We further observed reduced lactate dehydrogenase activity, low lactate concentrations, and failure of lactate-mediated lipolysis inhibition during exercise. Consequently, peripheral fats were mobilized to serve energy production in the muscle, as evidenced by the increased plasma lipid concentrations and increased transcript levels of enzymes of β-oxidation and the citrate cycle. Therefore, we hypothesized that lactate dehydrogenase activity significantly affects metabolic flexibility in DUhTP mice. 

Various strategies are pursued to assess and measure metabolic flexibility. In recent years, measuring the maximal FATox rate during a submaximal exercise intensity test has gained acceptance as both metabolic rate and substrate availability in tissues increase [27]. Other authors suggest the measurement of blood lactate alone as an effective method for measuring metabolic flexibility during exercise, as consistent and robust inverse correlations between blood lactate and FATox rate, obtained by indirect calorimetry, and between fat and carbohydrate oxidation rates, have been demonstrated in professional endurance athletes, moderately active individuals and patients with metabolic syndrome [28]. Others recommend the respiratory quotient (RQ) as a more sensitive marker of metabolic flexibility, especially in female subjects, than circulating lactate because of (i) its high variance in response to exercise in cohorts with different metabolic health and aerobic status and (ii) different management of cellular lactate production and excretion in women than men [29].

To test the hypothesis that marathon mice DUhTP are metabolically more flexible than the control line DUC, we now investigated whether we could corroborate this hypothesis by measuring lactate values or by indirect calorimetry. We collected these data three days after three weeks of training to examine the persisting effects of exercise on metabolic flexibility.

## 2. Materials and Methods

### 2.1. Animals and Treadmill Training

In this study, we used the non-inbred mouse line DUhTP, which was paternally selected for 140 generations for high treadmill performance [30], and the corresponding control line (DUC), which was generated from the identical base population without phenotype selection [31]. Animals of generations 151 (DUhTP) and 196 (DUC) were used. All animal experiments were performed in the FBN Lab Animal Facility in Dummerstorf, Germany. The experiments were approved by the internal institutional review board and were conducted following national and international animal welfare guidelines (Animal Welfare Act (TierSchG); AZ 7221.3−1−064/19). The animals were kept under specific pathogen-free (SPF) conditions in polysulfone cages of 267 × 207 × 140 mm (H-Temp PSU, Type II, Eurostandard Tecniplast, Hohenpreißenberg, Germany) in compliance with hygiene management and health surveillance according to FELASA recommendations. The housing conditions of the mice were defined by a 12 h light-dark cycle (room temperature = 22.5 °C ± 0.2 °C, humidity = 45%−60%). The animals were given autoclaved Ssniff^®^ M-Z food (12 kJ% fat, 27 kJ% protein, 61 kJ% carbohydrates; gross energy 16.7 MJ/kg; Ssniff-Spezialdiäten GmbH, Soest, Germany) and water ad libitum. All in all, the experimental conditions in the mouse model matched those in the previous study [17].

Also, as described in the preceding study [17], 52 male animals of both lines were kept in individual cages from the age of 49 days; one part of the animals (*n* = 16) completed a three-week treadmill program (“trained”) on a computer-controlled treadmill (TSE company), while the remaining animals (*n* = 10) were kept in their cages without training and served as a control group (“sedentary”, “sed”). Notably, both mouse lines are characterized by different running capacities [26]. To apply similar training intensities in both lines, male DUhTP and DUC animals trained for 30 min or 15 min per training unit, respectively, corresponding to about 23% of the average submaximal running duration in the previous generations (DUhTP: 133 min, DUC: 66 min, [26]) [17]. After an initial run and two-day break, regular training (5 x per week, 15/30 min; see Figure 1) was started. For this purpose, mice were exposed to an initial speed of 0.2 m/s for 20 s, then to a speed of 0.25 m/s (week 1), 0.33 m/s (week 2), 0.36 m/s (week 3) for 160 s, and finally to a speed of 0.33 m/s (week 1), 0.42 m/s (week 2), and 0.5 m/s (week 3) (Figure 1). The latter velocities were maintained for 1620 s in the DUhTP mice and 720 s in the DUC mice. As already mentioned in the previous work [17], during the experiment, it was noted that the DUC animals could not accomplish the final speed of 0.5 m/s. Hence, the animals remained at 0.42 m/s for the last week. The training was, therefore, adapted to the animals’ abilities based on their genetics. After 30 min, this corresponds to a completed distance of about 830 m; after 15 min, about 330 m. An experienced technician monitored the entire training run. A slider above the treadmill spatially limited the treadmill; the shock grid was not used. If an animal could not run, the run was stopped for this animal. If the same animal did not run the next day, it was excluded from the experiment. This was the case for two DUhTP and three DUC mice.

After the experiment, 70-day-old animals were sacrificed; plasma or serum was collected and stored at −20 °C [17]. Various tissues (liver, rectus femoris muscle, pituitary gland, and posterior subcutaneous fat) were removed, weighed, shock-frozen in liquid nitrogen, and stored at −70 °C and used for different studies, as previously published [17,32,33].

**Figure 1 cells-13-02123-f001:**
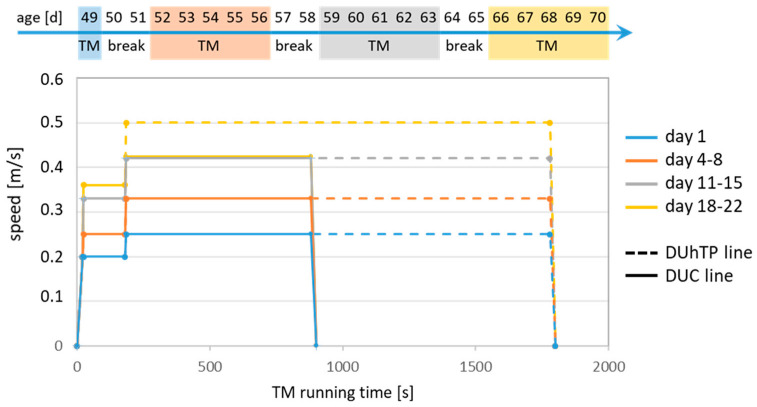
Three-week treadmill (TM) running protocol for DUhTP (dashed line) and DUC mice (unbroken line). Speed was increased weekly as indicated. The duration was adapted to the last submaximal running time achieved and corresponded to 23% of the respective average running performance (DUhTP: 30 min, DUC: 15 min; described in more detail in materials and methods and before [33]).

### 2.2. Metabolic Cages and Sample Collection

For this present study, however, a total of eighteen trained and sedentary animals of both lines (DUhTP sed: *n* = 5, DUhTP trained: *n* = 4, DUC sed: *n* = 5, DUC trained: *n* = 4 mice) were not sacrificed but transferred to metabolic cages (Panlab/Harvard Instruments, Holliston, MA, USA) with water and food ad libitum. After a one-day adaption period, metabolic data on oxygen consumption, carbon dioxide production, activity, feed, and water intake were collected over 48 h (two light and two dark periods). After the data recording in the metabolic cages, plasma samples were collected and stored at −20 °C for subsequent lactate measurements.

### 2.3. Lactate Measurements

Lactate was measured in plasma samples as recently described [17], using a commercial kit for L-lactate concentrations (A11A01721, Horiba, Kyoto, Japan).

### 2.4. Calculation of FATox and CHOox

Fatty acid oxidation and carbohydrate oxidation rates were calculated according to Frayn’s publication [34]. Therefore, the data on oxygen consumption and carbon dioxide production from the animals in each group during the 48 h observation period in the metabolic cage were used to calculate FATox and CHOox, neglecting nitrogen excretion and applying the formula FATox rates (g × min^−1^) = 1.67 VO_2_ (L × min^−1^) − 1.67 VCO_2_ (L × min^−1^) and CHOox rates (g × min^−1^) = 4.55 VO_2_ (L × min^−1^) – 3.21 VCO_2_ (L × min^−1^). The average values were determined. Oxidation rates of sedentary animals were set at 100%, and those of trained animals were set in relation to sedentary mice from the respective genetic groups. 

### 2.5. Statistical Analysis

Data analysis and graphical representations were performed using GraphPad Prism 9.1.0 (GraphPad Software, San Diego, CA, USA). The data were analyzed using two-way ANOVA with multiple comparisons. The effects and differences were considered significant if *p* < 0.05. Welch’s *t*-test was applied to compare FATox and CHOox of trained DUhTP and DUC mice.

## 3. Results

### 3.1. Total Body Metabolism in Response to Selection and Training

Oxygen consumption and carbon dioxide production were measured in trained (*n* = 4) and sedentary animals (*n* = 5) of both lines using metabolic cages. The observation period of 48 h included two light and two dark cycles of 12 h each (Figure 1). A significant increase in oxygen consumption and carbon dioxide production was observed during the dark periods compared to the light cycles in all groups (*p* < 0.0001; Figure 2a,b). In sedentary conditions, DUhTP mice displayed a lower oxygen consumption than DUC mice, especially during the second half of the light and the first half of the dark periods (each: *p* < 0.001; Figure 2a). On average, oxygen consumption during the light and dark periods differed only slightly by −8.7% and −5.8% (Figure 2c), respectively, which can be attributed to the temporal shift between DUhTP and DUC mice. Simultaneously, DUhTP mice showed higher carbon dioxide production during the second half of the light and the first half of the dark periods (Figure 2b). In general, sedentary DUhTP males showed, on average, 6.7% and 7.8% higher carbon dioxide production during the light and dark cycle, respectively, than sedentary DUC mice (Figure 2d).

As a result of training, the oxygen consumption of the DUC mice increased slightly by 9% and 11% during the light and dark cycles (Figure 2c). The difference in carbon dioxide production between trained and untrained DUC mice was about +23% during both cycles (*p* < 0.001, Figure 2d). In DUhTP mice, training increased oxygen consumption during hours with light (14%) and without light (24%, *p* < 0.0001; Figure 2c) and elevated carbon dioxide production during the dark phase (Figure 2d, 23%, *p* < 0.0001) compared to sedentary siblings.

Overall, trained DUhTP mice produced more carbon dioxide and consumed more oxygen than trained DUC animals during the dark cycles (*p* < 0.0001).

Consequently, the respiratory quotient (RQ), which displays the ratio between expired carbon dioxide and consumed oxygen, was the highest in sedentary DUhTP mice and corresponded to around 1 (Figure 3a,c, *p* < 0.0001). However, in control mice, the RQ at rest was about 0.81 (light cycle) and 0.87 (dark cycle). RQ was generally higher in dark periods, the active periods for nocturnal mice (*p* < 0.0001). In response to training, the average RQ decreased by around 7.4% in DUhTP mice during the light cycles (Figure 3c, *p* = 0.059) and increased by 9% and 9.3% in DUC mice during the light and dark cycles (*p* < 0.05), respectively. 

The energy expenditure of sedentary DUhTP and DUC mice was comparable during light and dark cycles. Notably, DUC mice showed a higher energy turnover in the first half of the dark phase, and DUhTP mice demonstrated a higher energy expenditure in the second half (*p* < 0.0001; Figure 3b). After training, both lines’ energy expenditures were higher than those of untrained littermates during both cycles, being statistically significant during dark cycles (*p* < 0.05; Figure 3d). However, energy turnover increase in response to training was higher in DUhTP (23%, *p* < 0.0001) than DUC mice (14%, *p* < 0.05). 

### 3.2. Increased Activity and Food Intake in Response to Training in DUhTP Mice

The average activity in the metabolic cages (Figure 4a) was higher during the dark cycles than the light ones, with trained DUhTP mice showing the greatest difference in activity between light and dark phases (4.7-fold, not significant). Total activity during the 48 h observation period increased by 55% with training in the DUhTP mice, while cage activity decreased by 19% between sedentary and trained DUC mice (both non-significant).

Food consumption during the dark cycles in the metabolic cages within 48 h increased significantly in DUhTP mice in response to training (Figure 4b, *p* < 0.01). It differed significantly from the intake of trained controls (*p* < 0.05). Sedentary and trained DUC mice showed a comparable calorie intake.

### 3.3. Reduced Lactate Concentration in Trained DUhTP Mice

As already published, only DUC mice showed significantly increased plasma lactate concentration directly after the training compared to sedentary DUC and trained DUhTP mice (both *p* < 0.05; Table 1 left). After three days (one day of acclimatization and two days of observation) in the metabolic cages, lactate concentrations of trained DUC mice resembled those of sedentary DUC mice (70 d and 73 d). Trained DUhTP mice instead had significantly lower lactate concentrations after three days of spending time in metabolic cages than trained littermates directly after training (*p* < 0.01; Table 1 right) or sedentary DUhTP mice (*p* < 0.05). Overall, the plasma lactate concentrations of the trained DUhTP mice from the metabolic cages tended to be lower than those of the trained DUC mice (−71%, *p* = 0.07).

### 3.4. Increased Fatty Acid and Carbohydrate Oxidation in Trained DUhTP Mice

The oxidation rates of fatty acids and carbohydrates were calculated with the data obtained from the metabolic cages regarding oxygen consumption and carbon dioxide production from the entire 48 h study period, neglecting nitrogen excretion, as is usual when calculating FATox or CHOox rates [29,35]. 

Trained DUhTP mice showed increased FATox (+33% and +40%; Figure 5a) and CHOox rates (+19.2% and +16.2%; Figure 5b) during light and dark cycles compared to their sedentary littermates. In contrast, trained DUC mice exhibited a lower FATox rate (−57.6% and −64.7%) and a higher CHOox rate (+21.6% and +17.5%) than the sedentary controls. FATox rates differed between trained mouse lines by 90.7% (*p* < 0.03) and 105.2% (not significant) during light and dark cycles.

## 4. Discussion

Our previous study [17] suggested that the marathon mouse line DUhTP responds metabolically more flexibly to high-speed training than the control line DUC. We attributed this assumption to increased transcript levels of encoding enzymes involved in CHOox and FATox found by next-generation sequencing in trained DUhTP mice compared to trained control mice. In this previous study [17], we proposed a crucial role for lactate dehydrogenase, as downregulation of its enzyme activity in trained DUhTP mice prevents increased lactate production during endurance exercise, allowing energy supply from peripheral fat depots [36,37], ensuring the high energy demand during endurance exercise is covered. Exercise-related metabolic flexibility is required to efficiently coordinate the considerable increase in energy demand during physical exercise with fuel availability and increased metabolism. To quantitatively assess metabolic flexibility, various methods, such as maximal FATox rate during a submaximal exercise intensity test, lactate determination, or respiratory quotient, have been proposed by different researchers [27,28,29]. We investigated whether we could prove our hypothesis of higher metabolic flexibility in trained DUhTP mice, as suggested in the previous study [17], by acquiring lactate concentrations and indirect calorimetry measurements after exercise, taking into account the effects of training and recovery.

### 4.1. Lactate as a Marker for Metabolic Flexibility

Lactate is a product of anaerobic glycolysis, and its blood concentration increases exponentially with higher exercise intensities. Highly trained athletes, instead, have lower blood lactate concentrations but higher average maximal oxygen uptake than non-athletes with the same relative training intensity [38,39]. In athletes who train at low intensity, it is postulated that a reduction in blood lactate concentrations may be a sign of efficient utilization of FATox and improved recovery during training [40]. Increased blood lactate concentrations are accompanied by higher glycolytic flux and a decreased FATox. A clear correlation between increased blood lactate concentrations and reduced FATox levels could be demonstrated, particularly the clear relationship between the onsets of the lactate concentration increase and the decrease in FATox [41]. Since San-Millán and Brooks also found robust inverse correlations between blood lactate and FATox, they postulated that measuring blood lactate alone is an effective way to indirectly assess mitochondrial function and metabolic flexibility during exercise [28].

We have previously found that DUhTP mice have lower lactate concentrations immediately after exercise than DUC mice and have linked this observation to reduced lactate dehydrogenase activity detected in trained DUhTP mice [17]. As lower lactate concentrations during exercise abrogate lactate-mediated lipolysis inhibition [37], and inverse correlations between blood lactate and FATox rate have been described [21,28], prolonged low blood lactate values may indicate a preference for oxidative metabolism in DUhTP mice in response to training.

Three days after the last exercise session, lactate concentrations in trained DUhTP mice were still lower than those of trained DUC mice as well as in DUhTP mice immediately after exercise or sedentary ones. These lower lactate concentrations in trained DUhTP mice suggest higher oxidation rates even three days after exercise and, thus, a lasting higher metabolic flexibility due to training.

### 4.2. RQ as a Marker for Metabolic Flexibility

According to Waldman and colleagues, the ratio of carbon dioxide production to oxygen consumption (RQ) is a more sensitive marker of metabolic flexibility than lactate, especially in female subjects with different oxidative capacities [29]. The authors found different substrate oxidation rates between obese and recreationally active or endurance-trained female subjects, resulting in significantly lower RQ than the overweight group. Lactate concentrations, instead, did not differ significantly. Only at a workload of 75 Watts did endurance-trained female subjects demonstrate significantly lower circulating lactate concentrations than overweight subjects. Therefore, the authors could demonstrate clear differences in RQ between overweight and exercising female subjects, pointing to higher metabolic flexibility in females with better metabolic and aerobic statuses. However, differences between recreationally active or endurance-trained females were not detectable.

Our data from metabolic cages obtained from both mouse lines, sedentary or physically active for three weeks, revealed (i) a higher RQ in sedentary DUhTP than in DUC mice and (ii) a decreasing RQ in DUhTP mice and an increasing RQ in DUC mice in response to training. 

During sedentary conditions, DUhTP mice consumed less oxygen but produced higher carbon dioxide than unselected controls, indicating increased lipid synthesis due to the release of CO_2_ during fatty acid elongation by two carbon atoms. The RQ, which represents the ratio of carbon dioxide produced to oxygen consumed and provides information on the substrates used as the predominant energy source, confirmed this assumption. An RQ clearly above 1, as observed in sedentary DUhTP animals and, for example, in fattening animals [42], indicates (i) carbohydrates as the predominant fuel source and (ii) the formation of depot fat from excess carbohydrates [34]. This correlates with previous work in which we demonstrated increased lipid synthesis in the liver of sedentary DUhTP mice [43]. An RQ of about 0.84, as observed in sedentary DUC mice, indicates the utilization of both carbohydrates and lipids for energy turnover and is consistent with results from other mouse studies [44,45].

In response to training, energy expenditure, oxygen consumption, and carbon dioxide production increased in both lines, indicating an increase in oxidative metabolism. A shift to higher carbohydrate consumption in DUC mice can be assumed because the RQ increased in response to training. For example, a comparable increase in RQ during exercise was observed in CD36 knockout mice [46]. FATox was reduced in these mice by inhibiting fatty acid uptake, and carbohydrates were used as the preferred substrate.

On the contrary, in DUhTP mice, RQ decreased due to training, indicating a higher preference for lipid degradation than in DUC mice. Exercise-mediated RQ reduction was also attributed to increased fat, but not glucose, oxidation in mice that underwent ten weeks of swimming training to exhaustion compared with untrained animals [47]. Also, 30-day treadmill training decreased RQ compared to untrained mice, suggesting fat as a preferred substrate during training [48]. Interestingly, RQ in DUhTP mice was reduced exclusively during light cycles in response to training, representing a training-induced basal metabolic rate with greater dependency on lipids than sedentary littermates. In the dark cycles, instead, RQ was still close to 1. Furthermore, trained DUhTP mice showed higher voluntary activity and increased food consumption during these active phases. This suggests that the training effect on lipid-based catabolism in DUhTP mice is maintained during inactive cycles and that selection-driven metabolism towards lipid synthesis and accumulation persists during the dark, active phases. The results of a previous study support this assumption, showing that DUhTP mice regain weight within three weeks after regular physical exercise and accumulate lipids in peripheral depots [49].

However, an RNA-seq data set derived from the muscle tissue of DUhTP mice directly after training compared to trained DUC mice demonstrated increased transcript levels of enzymes involved in glycolysis, the TCA cycle, fatty acid degradation, and oxidative phosphorylation [17]. Accordingly, at least in muscle, oxidative utilization of glucose and fatty acids can be assumed in response to training. In addition, higher activities of pyruvate dehydrogenase and isocitrate dehydrogenase and decreased lactate dehydrogenase activity were identified, indicating oxidative utilization of carbohydrates and fatty acids and, therefore, higher metabolic flexibility in response to training.

### 4.3. FATox as a Marker of Metabolic Flexibility

In healthy, moderately trained men, maximum FATox was assessed by a cycling protocol with 5 min stages and 35 Watt increments in the work rate [50], finding that FATox rates are high over a wide range of intensities, with a maximum of 0.6 at a workload of 64 ± 4% VO_2 max_. FATox rates decrease significantly at training intensities above maximum FATox, and above 89 ± 3% VO_2 max_, FATox hardly occurs [50]. However, FATox rates depend on macronutrient availability, training status, gender, training intensity, and exercise duration [51]. These factors can cause cellular adaptations influencing systematic fatty acid transport and, thus, FATox. An incremental 10,000 m treadmill test to estimate the individual maximal FATox rate based on gas measurements in trained amateur competitors in regional or national 10,000 m races tended to show higher maximal FATox in the moderate-performance group compared to the low-performance group [35]. However, as the maximal FATox rate was lower than in other experiments, such as uphill walking [52], the authors speculated that a greater contribution of CHOox and lower FATox to energy expenditure was necessary during running compared to walking or cycling [35]. A comparison of athletes and a general population completing a session of graded incremental exercise testing provided higher lactate concentrations and CHOox rates in the general population and a higher FATox rate in athletes [21]. Nevertheless, the oxidative energy system in both groups is predominantly used, and lactate and FATox correlate negatively.

These data or protocols cannot be adopted one-to-one since we did not measure oxygen consumption and carbon dioxide production during running in our experiment. Moreover, our study aimed to determine the exercise-associated changes in FATox and CHOox rates after three weeks of high-speed training. Furthermore, our animals were not subjected to a fasting period of 10 to 12 h, unlike in the approach by Achten et al. [50]. Instead, they were given a carbohydrate-containing mixed diet, recognized for its effect on the FATox rate [53]. We can show training-induced adaptations in FATox rates, indicated by increased FATox rates in trained DUhTP mice and decreased FATox rates in trained DUC mice compared to their respective sedentary littermates. CHOox rates increase in both lines to a similar extent. Our calculations regarding FATox and CHOox provided in this study suggest less metabolic flexibility with higher dependency on carbohydrates for energy production in trained DUC mice due to their increased CHOox but reduced FATox rates. Trained DUhTP mice, instead, show simultaneously increased oxidation of carbohydrates and fatty acids compared to sedentary DUhTP mice, which can be attributed to increased metabolic flexibility, even three days after training.

## 5. Conclusions

This study provides clear evidence that metabolic flexibility is elevated in male DUhTP mice, even three days after three weeks of training, compared to sedentary siblings. We used RQ and FATox as standard parameters to assess training-associated metabolic flexibility but also circulating blood lactate concentrations. All three parameters indicate a higher metabolic capacity in trained DUhTP than in trained DUC mice, as previously suggested in analyses based on RNA and protein levels [13]. Compared to their sedentary counterparts, trained DUhTP males showed a higher FATox rate, a lower basal RQ, and lower lactate concentrations. Trained DUC mice, on the other hand, showed a reduced FATox rate and a higher RQ than their sedentary littermates, as well as higher lactate concentrations than trained DUhTP mice. This indicates a shift to oxidative metabolic pathways in DUhTP mice and glycolytic metabolic pathways in DUC mice.

Accordingly, we can hypothesize that DUhTP mice fulfill energy demands during high-speed exercise via oxidative pathways. Thus, the elevated metabolic flexibility is linked to the improved running performance in DUhTP mice acquired by phenotype selection over more than 140 generations of breeding. The genetic tools of the DUhTP mouse line, as a result of long-term selection, are the driving force for the metabolic flexibility in this mouse model. The dependence of metabolic flexibility on genetic predisposition could explain similar mechanisms in other species, including humans.

## Figures and Tables

**Figure 2 cells-13-02123-f002:**
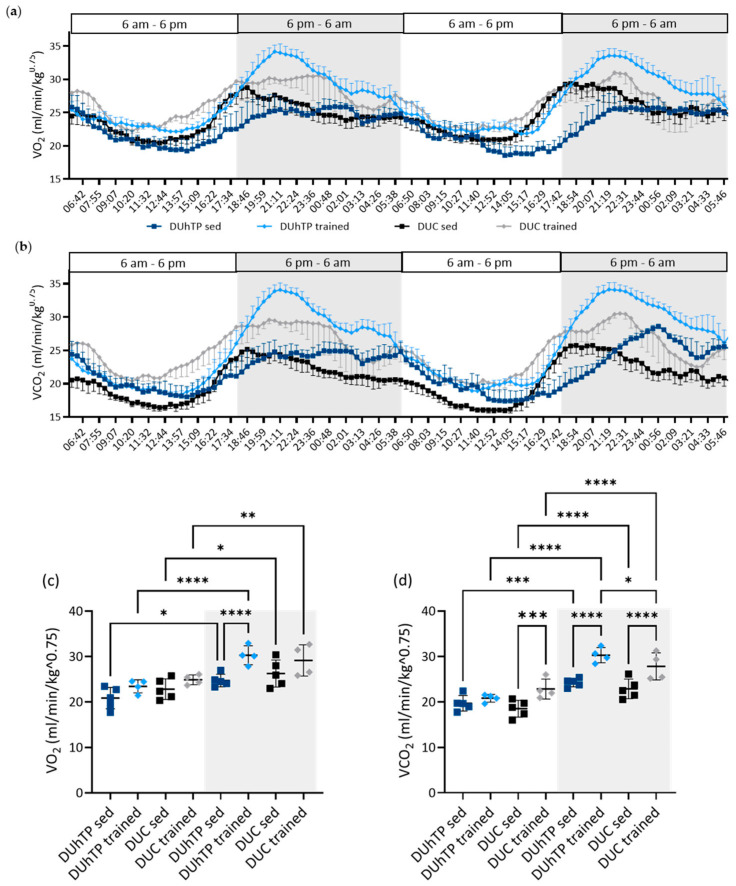
Oxygen consumption and carbon dioxide production were obtained for 48 h from trained and untrained DUhTP (light (*n* = 4 animals)/dark blue (*n* = 5 animals)) and DUC mice (gray (*n* = 4 animals)/black (*n* = 5 animals)) kept in metabolic cages. Diagrams (**a**,**b**) summarize the individual results of two 12 h light and two 12 h dark (gray-shaded) cycles, showing mean +/− standard error of mean. Diagrams (**c**,**d**) show the average oxygen consumption and carbon dioxide production during the light and dark cycles with standard derivation. Each dot represents the data of one animal. Statistical analysis was performed using two-way ANOVA. Significant differences as indicated: * *p* < 0.05; ** *p* < 0.01, *** *p* < 0.001, **** *p* < 0.0001.

**Figure 3 cells-13-02123-f003:**
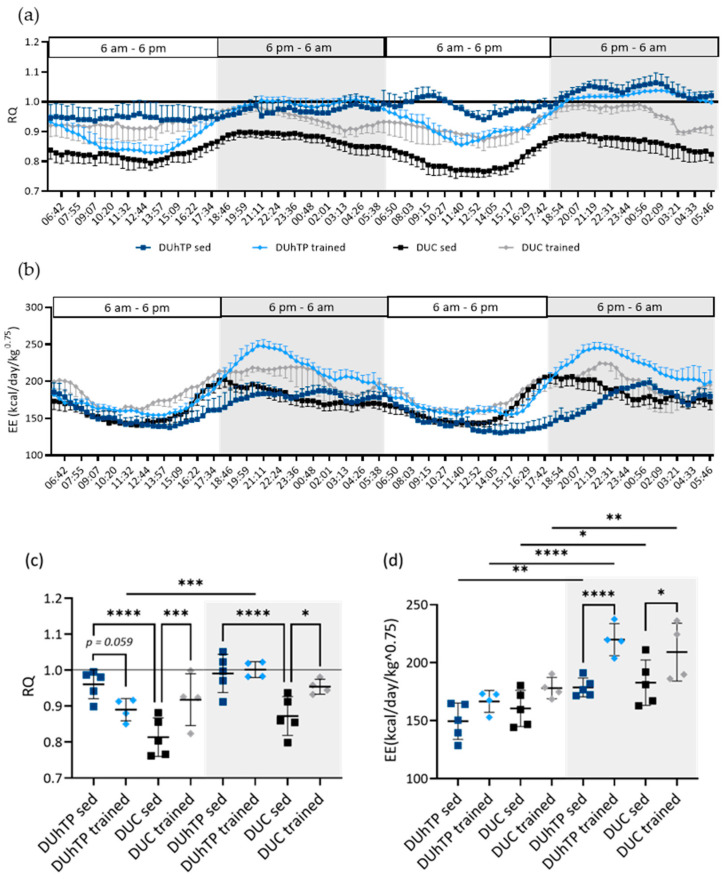
Respiratory quotient (RQ) and energy expenditures (EE) obtained for 48 h from trained (*n* = 4 animals) and untrained (*n* = 5 animals) DUhTP (light/dark blue) and DUC mice (gray/black), kept in metabolic cages. Results of two 12 h light and two 12 h dark (gray-shaded) cycles are shown as (**a**,**b**) means +/− SEM over time and of (**c**) average RQ and (**d**) EE during light and dark (gray-shaded) cycles are shown as scatter plots with mean and standard deviations; each dot represents one animal. Statistical analysis was performed using two-way ANOVA. Significant differences as indicated: * *p* < 0.05; ** *p* < 0.01, *** *p* < 0.001, **** *p* < 0.0001.

**Figure 4 cells-13-02123-f004:**
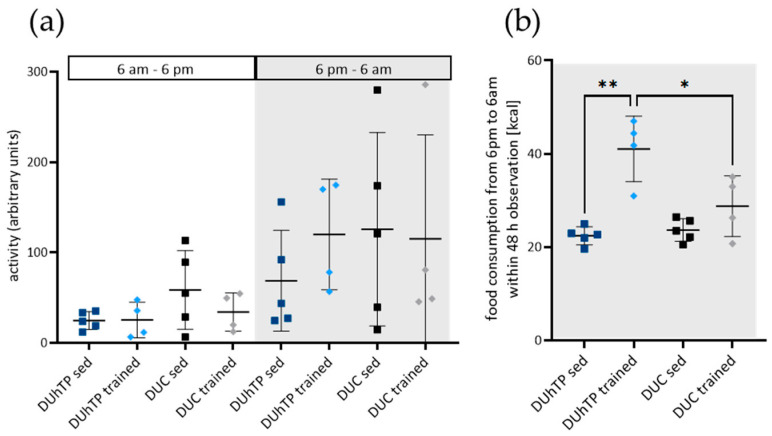
Cage activity and food consumption within 48 h were obtained in trained and untrained DUhTP (light (*n* = 4 animals)/dark blue (*n* = 5 animals)) and DUC mice (gray (*n* = 4 animals)/black (*n* = 5 animals)) kept in metabolic cages. Diagram (**a**) summarizes the individual activity of two 12 h light and two 12 h dark (gray-shaded) cycles, and diagram (**b**) represents the food consumption in kcal during the two dark cycles from 6 pm to 6 am. All results are presented as scatter plots with mean and standard derivation. Statistical analysis was performed using two-way ANOVA. Significant differences as indicated: * *p* < 0.05; ** *p* < 0.01.

**Figure 5 cells-13-02123-f005:**
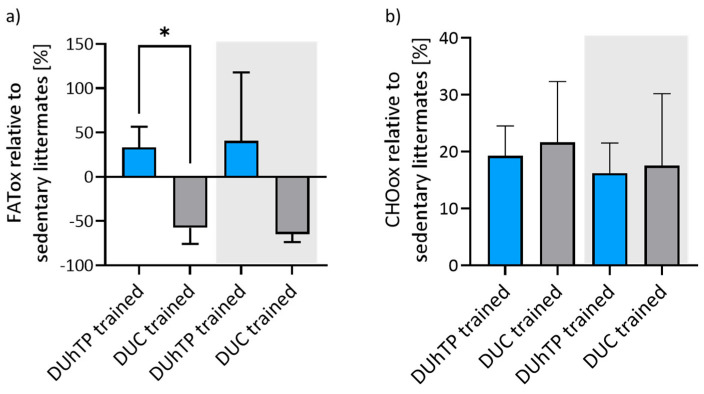
(**a**) Fatty acid oxidation (FATox) and (**b**) carbohydrate oxidation (CHOox) rate of trained (*n* = 4 animals) and untrained (*n* = 5 animals) DUhTP (light/dark blue) and DUC mice (gray/black). Oxidation rates were calculated with data obtained from metabolic cages and were represented in relation to the rates from their sedentary controls [100%]. Results are presented as bars with standard derivation. Statistical analysis was performed using Welch’s *t*-test. * *p* < 0.05.

**Table 1 cells-13-02123-t001:** Lactate concentration in mmol/L, measured in plasma of sedentary and trained DUhTP and DUC mice (controls) directly after training (left) and after three days in the metabolic cage (right). Plasma lactate data from littermates directly after training have already been published in [17]. Data are reported as mean +/− SD, and *n* is the number of mice analyzed. Statistical analysis was performed using two-way ANOVA. Significant differences as indicated: § vs. DUhTP trained (70 d) and vs. DUhTP sed (73 d); # vs. DUC sed (70 d) and DUhTP trained (70 d).

	Plasma Lactate Concentration [mmol/L]					
	In Siblings Directly After Training (Day 70); Published in [17]			After Observation in Metabolic Cages (73 Days Old)		
groups	mean	SD	*n*	mean	SD	*n*
DUhTP sed	4.96	0.45	8	4.89	0.33	5
DUhTP trained	5.40	1.30	12	2.78 ^§^	0.42	4
DUC sed	4.77	1.58	6	4.58	0.47	5
DUC trained	6.76 ^#^	1.22	11	4.67	1.02	4

## Data Availability

All raw data were generated at the FBN Dummerstorf. Derived data supporting the findings of this study are available on request from the corresponding author.

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
