# Peer review of "Lower Plasma Lactate Concentrations After Training Support the Hypothesis of Improved Metabolic Flexibility in Male Long-Term Selected Marathon Mice Compared to Unselected Controls"

_cells, 2024, doi:10.3390/cells13242123_

Round 1

Reviewer 1 Report

Comments and Suggestions for Authors

Title- please change the Title to reflect on mice, must be improved for clarity and conciseness. The title of your manuscript should be concise, specific and relevant. It should identify if the study reports (human or animal) trial data, or is a systematic review, meta-analysis or replication study. https://www.mdpi.com/journal/cells/instructions#preparation Please, write the Title in a way that it informs of the study outcome. Overall remarks of the manuscript: There are many papers on metabolic flexability. The main message of the manuscript is that the selected DUhTP mice, which exhibit high treadmill performance, demonstrate superior metabolic flexibility compared to the unselected DUC mice. This enhanced metabolic flexibility is demonstrated by their improved capacity to alternate between different fuels (fats and carbohydrates) following a training regimen. The research indicates that the trained DUhTP mice have a lower respiratory quotient, higher rates of fatty acid oxidation, and reduced blood lactate concentrations in comparison to their untrained counterparts, indicating a transition towards more efficient oxidative metabolism. Overall, these results imply that increased metabolic flexibility is linked with enhanced running performance in the DUhTP mouse strain. However, the current manuscript presents intriguing findings regarding the metabolic flexibility of the DUhTP mice, especially in light of their long history of paternal selection for high treadmill performance. While the DUhTP line was indeed paternally selected for high treadmill performance, this does not automatically imply enhanced metabolic flexibility. High performance could theoretically arise from several mechanisms. This has implications not just for mouse models but potentially for understanding similar mechanisms in other species, including humans. Please, make the manuscript to convey this message. Otherwise, the novelty should have come across more. To address concerns about novelty and whether the findings were expected, emphasize that although treadmill performance was specifically chosen for evaluation, the underlying metabolic adaptations contributing to this enhancement were previously uncertain, compare DUhTP to other endurance models, in both the Introduction and the Discussion, recognize the existing literature on metabolic flexibility, while stressing that the current study uniquely connects it to prolonged paternal selection for endurance, supported by experimental evidence.

Abstract

Please do not use „wondered” in scientific language, line 18

Please do not use „so” in scientific -line 26

„lactate levels”- I would change to „lactate concentrations”-line 17,23,25. The Abstract is self-sufficient and provides a clear overview of the study. The work is aimed at determining whether increased metabolic flexibility can be observed in DUhTP mice three days after training, particularly by comparing their responses to a high-speed treadmill training program with those of DUC mice. However, what DUhTP and DUC mice are needs to be clarified for the first-time reader. Please explain briefly. Please, adhere to the following: Acronyms/Abbreviations/Initialisms should be defined the first time they appear in each of three sections: the abstract; the main text; the first figure or table. https://www.mdpi.com/journal/cells/instructions#preparation

Please revise the Abstract as should be a total of about 200 words maximum.

1.Introduction

Please delete „respective” in line 57.

Add a comma before „and oxidation’ in line 61.

In scientific writing, "small" and "large" should be used cautiously, as they are relative and can lead to ambiguity. Lines 36, 84, respectively.

Why exactly three days after training? Line 91.

Lines 37,38, 70,72- here are the „levels” again. Please revise.

„wanted” in line 89. Please replace with other verb or rewrite the sentence.

The Introduction clearly outlines the justification for the work by addressing the current understanding of metabolic flexibility and its significance in diet and exercise. It emphasizes essential ideas, including the capacity to alternate between oxidative fuel sources, the physiological effects of metabolic flexibility during exercise, and the adverse health impacts of metabolic inflexibility. The Introduction outlines an interesting objective for the study, i.e., to explore the metabolic flexibility of DUhTP mice compared to DUC mice, particularly concerning training. The hypothesis proposes that DUhTP mice may exhibit enhanced metabolic flexibility due to superior running endurance. However, the Introduction would benefit from more explicitly articulating the specific objectives or research questions to be addressed in the study. Clarifying these aims would provide greater clarity and focus for the investigation, helping readers better understand the study's goals within metabolic flexibility and related concepts. More context regarding the implications of metabolic inflexibility in health would be beneficial. Instead of solely referencing obesity and type 2 diabetes, it could benefit from more specific examples or clinical implications of impaired metabolic flexibility. How would metabolic flexibility relate to sports?

2. Materials and Methods

2.1 Animals and Experimental Design

„All animal experiments were performed in the Lab Animal Facility of the FBN in Dummerstorf, Germany.” Lines 98-100. Please, replace with: „All animal experiments were performed in the FBN Lab Animal Facility in Dummerstorf, Germany”-if this is what was meant.

Please delete „so” in line 124. See my previous comment regarding the use of „so”-Abstract. Rewrite the sentence „As already mentioned in the previous work [13], during the

experiment, it was noted that the DUC animals could not accomplish the final speed of 0.5 m/s, so the animals remained at 0.42 m/s for the last week.” Lines 123-125.

What is "SPF barier”? Line 102/103.

2.5 Statistical Analysis

„The data were analyzed using two-way ANOVA, multiple comparisons.” Please rewrite this to have: „The data were analyzed using two-way ANOVA and multiple comparisons.”

General remarks on Materials and Methods

The research encompasses both trained and sedentary groups of DUhTP and DUC mice, facilitating direct comparison of metabolic flexibility between these two mouse lines across varying physical conditions.

A three-week high-intensity treadmill training regimen allows for adequate evaluation of the training's effect on metabolic parameters. The study includes an acclimatization day, ensuring the mice are settled before measurements start.

The description does not mention the sample size for each experimental group. The training duration is mentioned. However, specific details concerning treadmill training sessions' intensity, duration, and frequency still need to be included.

3. Results

„Looking at the overall activity during the 48-hour observation period, the training increased the cage activity of the DUhTP mice by 55 %, while it decreased by 19 % in the DUC mice (both not significant).” Lines 239-241. This is poorly written. Please, revise.

„cycles, and”-remove the comma here in line 252.

General remarks Results:

Was it intentional to say „standard derivation”?-line 205 Why is sometimes SEM, and sometimes SD used? Why not to stick to one?

4. Discussion

„We wanted” please delete and replace with something else, line 303/304

„RQ as marker for metabolic flexibility” Please add „a” before „marker” line 333

„FATox as marker of metabolic flexibility”-please add „a” before „marker” in line 389

„In addition, our animals were not fasted for 10 to 12 hours like done by Achten et al. [38] but were fed a carbohydrate-containing mixed diet, which is known to affect the FATox rate [41].” This sentense is confusing. What does it mean „not fasted”? Lines 411-413. Perhaps this is what was meant: „Unlike the approach taken by Achten et al. [38], our animals were not subjected to fasting period of 10 to 12 hours. Instead, they were given a carbohydrate-containing mixed diet, which is recognized for its effect on the FATox rate [41].

„These lower lactate levels in trained DUhTP mice suggest higher oxidation rates even three days after training and thus a long-lasting higher metabolic flexibility.” Please replace with:

There are many words „leve/levels”. Please replace tchem with other words. See my previous comments about „level”, please.

5. Conclusions There are many words „levels”. Please replace them with other words. See my previous comments about „level”, please.

Author Response

Dear Reviewer 1,
Thank you for your time and effort in reviewing our manuscript. We thank you for your appreciative words about our work and the extremely helpful comments, which contributed significantly to improving our manuscript. Thank you!

Reviewer 1:

R: Title- please change the Title to reflect on mice, must be improved for clarity and conciseness. The title of your manuscript should be concise, specific and elevant.

It should identify if the study reports (human or animal) trial data, or is a systematic review, meta-analysis or replication study.

https://www.mdpi.com/journal/cells/instructions#preparation

Please, write the Title in a way that it informs of the study outcome.

Thanks for these helpful remarks. We adjusted the title accordingly.

R: Overall remarks of the manuscript: There are many papers on metabolic flexability. The main message of the manuscript is that the selected DUhTP mice, which exhibit high treadmill performance, demonstrate superior metabolic flexibility compared to the unselected DUC mice. This enhanced metabolic flexibility is demonstrated by their improved capacity to alternate between different fuels (fats and carbohydrates) following a training regimen. The research indicates that the trained DUhTP mice have a lower respiratory quotient, higher rates of fatty acid oxidation, and reduced blood lactate concentrations in comparison to their untrained counterparts, indicating a transition towards more efficient oxidative metabolism. Overall, these results imply that increased metabolic flexibility is linked with enhanced running performance in the DUhTP mouse strain. However, the current manuscript presents intriguing findings regarding the metabolic flexibility of the DUhTP mice, especially in light of their long history of paternal selection for high treadmill performance. While the DUhTP line was indeed paternally selected for high treadmill performance, this does not automatically imply enhanced metabolic flexibility. High performance could theoretically arise from several mechanisms. This has implications not just for mouse models but potentially for understanding similar mechanisms in other species, including humans. Please, make the manuscript to convey this message. Otherwise, the novelty should have come across more. To address concerns about novelty and whether the findings were expected, emphasize that although treadmill performance was specifically chosen for evaluation, the underlying metabolic adaptations contributing to this enhancement were previously uncertain, compare DUhTP to other endurance models, in both the Introduction and the Discussion, recognize the existing literature on metabolic flexibility, while stressing that the current study uniquely connects it to prolonged paternal selection for endurance, supported by experimental evidence.

We would like to thank the Reviewer for these appreciative words regarding our work. Furthermore, we would like to thank the Reviewer for the advice on emphasizing the special features of our findings, which we have included in the discussion and in the conclusion.

Abstract

R: Please do not use „wondered” in scientific language, line 18

Thanks for this hint. We replaced „wondered“ by „investigated“ .

R: Please do not use „so” in scientific -line 26

We have deleted the word.

R: „lactate levels”- I would change to „lactate concentrations”-line 17,23,25.

Thanks for this correction. We replaced all „lactate levels” by „lactate concentrations”.

R: The Abstract is self-sufficient and provides a clear overview of the study. The work is aimed at determining whether increased metabolic flexibility can be observed in DUhTP mice three days after training, particularly by comparing their responses to a high-speed treadmill training program with those of DUC mice. However, what DUhTP and DUC mice are needs to be clarified for the first-time reader. Please explain briefly. Please, adhere to the following: Acronyms/Abbreviations/Initialisms should be defined the first time they appear in each of three sections: the abstract; the main text; the first figure or table. https://www.mdpi.com/journal/cells/instructions#preparation Please revise the Abstract as should be a total of about 200 words maximum.

We revised the abstract accordingly and shortened it to less than 200 words.

1.Introduction

R: Please delete „respective” in line 57.

Is deleted.

R: Add a comma before „and oxidation’ in line 61.

Is added.

R: In scientific writing, "small" and "large" should be used cautiously, as they are relative and can lead to ambiguity. Lines 36, 84, respectively.

Thanks for theses hints. The words are replaced by “slight” and “high”. (lines 46, 119)

R: Why exactly three days after training? Line 91.

Thank you for this question. The idea of a three-day study period is related to the study design. On the day of the last training session, after five consecutive days of running, we placed the mice in the metabolic cages. On the morning of the next day, the metabolic monitoring began.

This goes hand in hand with the study design in which five training days were followed by a two-day training break.

R: Lines 37,38, 70,72- here are the „levels” again. Please revise.

We replaced all “lactate levels” by “lactate concentrations”.

R: „wanted” in line 89. Please replace with other verb or rewrite the sentence.

We deleted the verb.

R: The Introduction clearly outlines the justification for the work by addressing the current understanding of metabolic flexibility and its significance in diet and exercise. It emphasizes essential ideas, including the capacity to alternate between oxidative fuel sources, the physiological effects of metabolic flexibility during exercise, and the adverse health impacts of metabolic inflexibility. The Introduction outlines an interesting objective for the study, i.e., to explore the metabolic flexibility of DUhTP mice compared to DUC mice, particularly concerning training. The hypothesis proposes that DUhTP mice may exhibit enhanced metabolic flexibility due to superior running endurance. However, the Introduction would benefit from more explicitly articulating the specific objectives or research questions to be addressed in the study. Clarifying these aims would provide greater clarity and focus for the investigation, helping readers better understand the study's goals within metabolic flexibility and related concepts.

More context regarding the implications of metabolic inflexibility in health would be beneficial. Instead of solely referencing obesity and type 2 diabetes, it could benefit from more specific examples or clinical implications of impaired metabolic flexibility. How would metabolic flexibility relate to sports?

Once again, many thanks for these appreciative words regarding our work. In the revised manuscript, we have implemented the notes on the specific objectives and provided more context regarding the implications of metabolic inflexibility/flexibility.

  1. Materials and Methods

2.1 Animals and Experimental Design

R: „All animal experiments were performed in the Lab Animal Facility of the FBN in Dummerstorf, Germany.” Lines 98-100. Please, replace with: „All animal experiments were performed in the FBN Lab Animal Facility in Dummerstorf, Germany”-if this is what was meant.

We changed the sentence accordingly. Thanks!

Please delete „so” in line 124. See my previous comment regarding the use of „so”-Abstract. Rewrite the sentence As already mentioned in the previous work [13], during the experiment, it was noted that the DUC animals could not accomplish the final speed of 0.5 m/s, so the animals remained at 0.42 m/s for the last week.Lines 123-125.

We changed the sentence accordingly. Thanks!

R: What is "SPF barier”? Line 102/103.

We apologize for this missleading term. We rewrote the sentence. It is now written: The animals were kept under specific pathogen-free (SPF) conditions in polysulfone cages of 267x207x140 mm (H-Temp PSU, Type II, Eurostandard Tecniplast, Hohenpreißenberg, Germany) in compliance with hygiene management and health surveillance according to FELASA recommendations. Lines 136-137

2.5 Statistical Analysis

R: „The data were analyzed using two-way ANOVA, multiple comparisons.” Please rewrite this to have: „The data were analyzed using two-way ANOVA and multiple comparisons.”

We changed the sentence accordingly. Thanks!

R: General remarks on Materials and Methods

The research encompasses both trained and sedentary groups of DUhTP and DUC mice, facilitating direct comparison of metabolic flexibility between these two mouse lines across varying physical conditions.

A three-week high-intensity treadmill training regimen allows for adequate evaluation of the training's effect on metabolic parameters.

The study includes an acclimatization day, ensuring the mice are settled before measurements start.

The description does not mention the sample size for each experimental group.

The training duration is mentioned. However, specific details concerning treadmill training sessions' intensity, duration, and frequency still need to be included.

We thank the reviewer for this comment, but are surprised as we believe all the information was included, at least in Fig. 1. However, as this is not clear enough, we have adapted the text and described the information on treadmill training sessions' intensity (lines 122-126), duration (lines 122-127), and frequency (line 122) in more detail. In addition, we have added the original number of animals used for the entire experiment, including the animals that dropped out. The number of animals that were examined in the metabolic cages can still be found in the lines 187-189.

  1. Results

R: „Looking at the overall activity during the 48-hour observation period, the training increased the cage activity of the DUhTP mice by 55 %, while it decreased by 19 % in the DUC mice (both not significant).” Lines 239-241. This is poorly written. Please, revise.

Thanks for pointing out this weakness. We changed this sentence accordingly. Lines 284-286

R: „cycles, and”-remove the comma here in line 252.

The comma is deleted.

General remarks Results:

R: Was it intentional to say „standard derivation”?-line 205 Why is sometimes SEM, and sometimes SD used? Why not to stick to one?

Thanks for this remark. All our results are normally represented by means with standard deviations. For the presentation of oxygen consumption, carbon dioxide production, respiratory quotient, and energy expenditure over 48 hours for all four groups in one diagram each, we have used the SEM and the standard error bar on only one side for a clearer presentation. We have presented these results mainly to show the observed temporal shift of the graphs between sedentary DUC and DUhTP mice. Nevertheless, all results were summarized separately into light and dark phases/cycles. These diagrams then contain the mean values with SD. However, I noticed that precisely these standard errors are missing in Fig. 3c. Therefore, the figure has been replaced.

  1. Discussion

R: „We wanted” please delete and replace with something else, line 303/304

The verb is deleted.

„RQ as marker for metabolic flexibility” Please add „a” before „marker” line 333

Is added.

R: „FATox as marker of metabolic flexibility”-please add „a” before „marker” in line 389

Is added.

R: „In addition, our animals were not fasted for 10 to 12 hours like done by Achten et al. [38] but were fed a carbohydrate-containing mixed diet, which is known to affect the FATox rate [41].” This sentense is confusing. What does it mean „not fasted”? Lines 411-413.

Perhaps this is what was meant: „Unlike the approach taken by Achten et al. [38], our animals were not subjected to fasting period of 10 to 12 hours. Instead, they were given a carbohydrate-containing mixed diet, which is recognized for its effect on the FATox rate [41].

We thank the Reviewer for drawing attention to our misleading sentence. We have implemented the Reviewer's suggestion accordingly.

R:„These lower lactate levels in trained DUhTP mice suggest higher oxidation rates even three days after training and thus a long-lasting higher metabolic flexibility.” Please replace with:

There are many words „level/levels”. Please replace tchem with other words. See my previous comments about „level”, please.

I have replaced all terms “lactate levels” with “lactate concentrations” as the Reviewer suggested. Thanks for this correction!

  1. Conclusions

R: There are many words „levels”. Please replace them with other words. See my previous comments about „level”, please.

They are replaced.

Reviewer 2 Report

Comments and Suggestions for Authors

Title. I suggest to replace “Suppression of” with “Lower”.

L29. Fat metabolism is always oxidative so I suggest to delete “oxidative” as carbohydrate metabolism has a glycolytic and oxidative part.

L36. Are you suggesting the insulin response is different in overweight or obese after intake of a high-fat diet. Please provide a reference to support that as Ref 2 does not.

L46. “physical activity” is considered any bodily movement. I suggest to replace by “exercise”.

L295. I suggest to replace “In this study” with “In our previous study [13],”

Ls 328-333. There seems to be surprise here but the mice were not examined after an extended detraining period. The adaptations by the physical training can be expected to return to pre-training levels but that could take more than 3 days. Please reconsider these statements.

Author Response

Dear Mr. Reviewer 2,
Thank you for your time and effort in reviewing our manuscript. We thank you for your helpful comments, which have contributed to the improvement of our manuscript.

Reviewer 2:

R: Title. I suggest to replace “Suppression of” with “Lower”.

We revised the title accordingly. Thanks!

R: L29. Fat metabolism is always oxidative so I suggest to delete “oxidative” as carbohydrate metabolism has a glycolytic and oxidative part.

We absolutely agree with the Reviewer. Since we shortened the abstract, we also deleted and revised this term. Thanks!

R: L36. Are you suggesting the insulin response is different in overweight or obese after intake of a high-fat diet. Please provide a reference to support that as Ref 2 does not.

As the Reviewer 1 wanted more examples about metabolic flexibility and further information regarding the clinical effects of impaired metabolic flexibility, we have listed further examples and discussed in that context the impact of a high-fat diet on insulin resistance / sensitivity. Lines 63-68

R: L46. “physical activity” is considered any bodily movement. I suggest to replace by “exercise”.

We revised the sentence accordingly. Thanks!

R: L295. I suggest to replace “In this study” with “In our previous study [13],”

We revised the sentence accordingly and moved the citation from the end to the beginning of the sentence. Thanks!

R: Ls 328-333. There seems to be surprise here but the mice were not examined after an extended detraining period. The adaptations by the physical training can be expected to return to pre-training levels but that could take more than 3 days. Please reconsider these statements.

Thank you for this comment. We agree with the Reviewer that our statement was misleading and that three days of training break is of course not a long-lasting effect that can be compared with 3 weeks of detraining. We have, therefore, revised the section that can be found in lines 433-438.